# Analysis Method for Post-Impact Damage Development in Carbon Fiber Reinforced Laminate under Repeated Loading

## Nikolai Turbin * and Kirill Shelkov

Moscow Aviation Institute (National Research University), Moscow 125080, Russia; shelkovka@mai.ru
* Correspondence: turbinnv@mai.ru

**Abstract:** In the current work, an analysis method for obtaining post-impact damage propagation under cyclic compressive load in flat carbon fiber reinforced plastic (CFRP) panels is presented. The solution for damage growth life is given based on the introduced hypothesis of reference damage mode (RDM). The critical size of damage for obtaining damage growth life was informed by the analysis of crack driving force versus damage size conducted using finite element analysis (FEA). The applicability of the damage tolerance principle for the case of compression–compression cyclic loading of the structural element containing impact damage is discussed and illustrated by the example. The results of using the introduced simplified approach to the calculation of characteristics of damage growth life suggest that the use of the slow-growth approach in composite structures is possible, though the necessity of obtaining the exact parameters of the damage growth rate equation with regard to the chosen crack driving force measure must be addressed.

**Keywords:** CFRP; delamination; fatigue after impact; barely visible impact damage; crack growth life; crack growth rate; damage tolerance

## 1. Introduction

The application of laminated polymer composite materials in primary structures of aircraft requires strength evaluation. Meanwhile, analysis methods for strength substantiation have evolved mainly in the field of static strength. This implies that assuring the long-term structural performance requirements relies mainly on testing. In particular, the lack of reliable methods for fatigue strength and damage tolerance estimates for composite structural elements leads to the implementation of the design principle, which restricts any progression of damage during the complete service objective (known as the no growth approach). Practically, this is achieved by using very conservative allowables for static strength [1]. In such a structure, the level of applied stresses from operational loads is well below the level sufficient for the strength or stiffness degradation of the material.

Relevant regulatory documentation [2], which formulates the aspects of compliance to safety requirements for the structure, in general, does not restrict the implementation of controlled (or slow-growth) criteria in treating the contained damage in the composite structure. For the implementation of the slow-growth principle in design and substantiation of the structural integrity, which, in this case, should lead to improved weight performance, the following is needed:

- The analysis method for damage characterization in terms of magnitude, size, and distribution in composite elements after a given impact event;
- The analysis method for damage accumulation in composite elements containing impact damage after the application of operational load cycles;
- Means for mapping the damage distribution obtained analytically to one obtained using non-destructive evaluation of the structure (NDE) and vice versa.

The derived characteristics of damage after applied load cycles, such as damage distribution, accumulation rate, and crack growth life, can readily be used in the elaboration

of structural inspection intervals and interpretation of damage patterns from NDE. Accurate analytical estimates of compression after impact (CAI) and fatigue after impact could be employed in the preliminary design routines such as lay-up and thickness of the laminate choice, which must reduce the weight penalty of the composite structure in terms of margins of safety.

It is well-known, that the presence of impact damage significantly compromises the strength of the composite structural element [3–13]; in particular, the compression strength reduction is pronounced—the so-called compression after impact property (CAI). Contemporary freighters such as MC-21, Boeing-787, Airbus-350, and others contain composite upper wing panels that work under compression loading. In this way, residual strength after an impact event during service is incorporated into design requirements and is validated by structural testing. The complexity of the composite structure performance under compression load as well as the phenomenon of the performance reduction of the composite element after impact and the practical necessity to control it through the design of an effective and safe structure explain the research interest in this question [6,7,9,10,12,14–16]. At the same time, a major part of the investigations and developed testing methods are dedicated to the question of static strength, leaving fatigue strength and damage tolerance aspects without a comparable amount of study both on theoretical and experimental levels [17,18].

Impact damage itself is characterized by the combination of different modes of material failure, distributed in the laminate plane and thickness [5], interacting with each other. Nevertheless, the generic appearance of impact damage modes is reproducible in experiments [3,4,7,9,10], such as fiber breakage in several layers on the impact site and in thin laminates on the bottom site; matrix cracking in the boundaries of the circumscribed cylinder about the contact point with impactor spread within significant thickness from the impact site; and delaminations of layers unevenly distributed through the thickness of the laminate. From the point of view of non-destructive evaluation (NDE) of the structure, such as ultrasound-based methods, the damage is found only in the form of delaminations; hence, only in-plane damage is recognized. This means that two other damage modes— fiber breakage and matrix cracking—can be evaluated only indirectly by the presence of localized delaminations of certain areas natured from them. Another obstacle for damage characterization by means used in structural maintenance non-destructive methods is obscuring of the delaminations and other damage modes under the delamination of the largest area [17], which leads to the detection of the largest delamination, but not the actual distribution of damage inside the structural element.

The definition of the damage measure is a must for the development and utilization of the methods for damage tolerance evaluation of the structure. To the authors' knowledge, up until now, there is no agreed measure of impact damage characterization, while in several studies [3,4,13,18] they use the area or length of the delamination. It must be noted that experimental curves for delamination propagation rate and size are obtained in the presence of other interacting modes of failure, which must be accounted for in analysis estimates. Moreover, the relevant damage measure will depend on the cyclic load conditions that the structural element is subjected to after the impact. In this way, it was investigated in the extensive and systematic experimental campaign by NASA [3,4] that in the case of symmetrical load cycles, block cycles, and quasi-random block cycles with dominating tensile load impact, damage propagates along the loading direction and starts from the free edges of the specimen, while in case of dominating compression cycles, delamination propagates from the impact site in a normal to loading direction and along the loading direction (which might be caused by the peculiarities of specimen fixture as the authors of [3,4] report.

Experimental evidence of impact damage specimen testing for fatigue strength [3,4,13,18] under compressive load cycles suggests that the damage propagates in the form of delamination of 2–3 layers of substrate under the surface, which occurs after its loss of stability (buckling). This peculiarity allows the investigation of fatigue after impact performance

using the so-called Instability Related Delamination Growth (IRDG) [19–23] theory for this exact type of delamination. Additionally, the prevalent framework for generic delamination analysis [24–26] requires the following data to be gained from the experiment: fracture toughness of the material for pure separating and sliding modes of delamination—$G_{Ic}, G_{IIc}$; energy release rate (ERR) $G$ as a function of delamination measure $a$—$G(a)$; rate of delamination measure propagation $\frac{da}{dN}$ ($N$—applied load cycles) as a function of relevant measure for cyclic damage ERR $G$ and cycle asymmetry—$\frac{da}{dN} = f(G, R)$, $R$—is a coefficient of load asymmetry, which must be the representative of the local ERR variation in the vicinity of the damage in contrast with external load variations during the fatigue cycle, which gives different values of $R$ [27]. It is worth mentioning, that the review of existing dependencies of type $\frac{da}{dN} = f(G, R)$ [25] does not present the universal measure of ERR $G$: the candidates are peak value of ERR $G$, range of ERR $\Delta G$, $\sqrt{G}$ and other variants, with an ability of a given type of ERR to reach a good approximation of test data in this exact case, but none of them for every case.

For the principle of slow-growth of damage to be used in the design of primary composite structures, the special condition must be met. As it follows from the influential works on the subject [24,27,28], such conditions, in the first place, are the gradual decrease in the crack-driving force (ERR) with the damage extension coupled with retardation of the damage propagation rate $\frac{da}{dN}$ after applied load cycles. For the practical use of the slow-growth approach method, it is also necessary to obtain the critical size of damage $a_f$ required for damage growth period evaluation. The latter, in turn, is used for service inspection interval detection. For the case of impact damage, the allowable damage size (i.e., critical size $a_f$) is not defined [17]. It must be added that for different locations of the structure, depending on their design and loading conditions, the allowable damage $a_f$ will be different.

In the present study, the method for the analysis of barely visible impact damage (BVID) growth in flat composite panels under cyclic compression load is described. At first, the hypothesis of the reference mode of damage is introduced, which allows the use of a conventional form of Paris-type damage propagation rate equation [29] for a single mode of damage while taking into account interacting modes of failure. Next, the criteria for damage tolerance of composite panels with BVID is suggested and used in the derivation of damage growth rate and damage growth period until the critical damage size is reached. The presented methodology is illustrated by an analysis example, which reveals the applicability of slow-growth ideology in the case of a composite structure subjected to cyclic compression.

## 2. Method

After considering all of the significant aspects of impact damage growth, mentioned in the Introduction, the main assumptions and analysis steps of the damage growth period model can be established. The core of the methodology is an author's hypothesis of reference mode of damage growth, which reads: it is assumed, that the impact damage growth driving process is progressive delamination under buckled substrate of the laminate and that the rate of the development of delamination is faster than the rate of development of other modes, mainly the rate of matrix cracks densification. Nevertheless, by knowing that delaminations and cracks will inevitably interact [30,31], it might be accepted that the two modes will develop in parallel to each other and have the same boundaries. At the same time, so that the damage zone for matrix cracking will gradually increase, the crack density in the damaged region will have the same value and hence the same stiffness loss within the region. Using the proposed hypothesis, the analysis steps for impact damage growth period estimation can be as follows:

1. The after-impact damage state of the composite is obtained. In light of the known BVID appearance and the independence of the cyclic damage growth methodology from impact damage distribution methodology, the typical form of BVID is accepted a priori and characterized by the inherent distribution of failure modes in the plane

and across the thickness of the laminate. A more detailed solution for impact damage mode distribution can be made by utilizing existing approaches [5–10], including numerical approaches based on finite element analysis (FEA). It is taken into account that the accepted initial size of damage must coincide with the definition of BVID, the threshold of which depends on the sensitivity of the chosen method of NDE.

2.  For thin (2–3 layers) substrate, delaminated and buckled under cyclic load levels, ERR modes are obtained for $G_I, G_{II}, G_{III}$. This part is conveniently accomplished using the virtual crack closure technique VCCT [32].

3.  Assuming that propagation of the damage will not affect its initial shape, the ERR change is found depending on the delamination size *a* (half of the delamination size normal to loading direction): $G(a)$. Obtained dependencies are approximated by trend lines and characteristic points are identified.

4.  The damage growth equation takes the standard form (1):

$$\frac{da}{dN} = cG_{equiv}^{\beta} \qquad (1)$$

where $G_{equiv}$—is an equivalent ERR, which combines in it three modes of delamination altogether; $c, \beta$—experimental constants for damage growth rate equation. For the cyclic compression case under consideration, $R = \infty$, the minimum external load and minimum ERR in the vicinity of the crack tip are both zero.

5.  The critical size of the damage is identified $a_f$. One of the options is such an extent of the damaged zone with reduced stiffness, for which the stress concentrations on the boundaries will cause the material to fail under the applied level of compression, i.e., exceed the compression strength of the intact material. The damaged zone with reduced stiffness is deduced by means of classical lamination theory (CLT) and replacement of the initial layered structure with an equivalent orthotropic single-layer plate, the elastic properties of the inclusion are obtained directly with stiffness reduction in each layer and using the rule of mixtures to obtain the stiffness degradation due to delamination [13,28]. The coefficient of stress concentration (SCF) in the orthotropic plate with elastic inclusion is obtained using known expressions [33] and finite-width correction [5]. In fact, the finite-width correction leads itself to the expression for the critical size of the damage. An alternate way is to decide the critical damage size with respect to damage detection thresholds of in-service NDE instruments and established design allowable. As an example, such a size might be the one taken for so-called visible impact damage (VID) or obvious damage, which are reliably recognized during inspections by means of instrumental NDE and the naked eye, respectively. In any case, the critical damage size is decided after the analysis of $G(a)$ dependency and stress–strain fields for each considered damage size, taken for $G(a)$ dependency development.

6.  Having the initial and final delamination size of the substrate zone and trends of $G(a)$ dependencies, and in light of the introduced hypothesis of reference damage mode, the damage growth period is obtained by integrating the equation of damage growth rate (1).

## 3. Method Usage

### 3.1. Virtual Crack Closure Technic (VCCT)

The energy release rate was determined using finite element models with VCCT (virtual crack closure technic)—a technique based on the principles of linear elastic fracture mechanics that allows modeling delamination between layers. This method uses the assumption that the energy released during crack growth is equal to the energy required to close the crack. Within the framework of the technique for spatial models, three delamination modes and the corresponding energy release rates are considered: mode I—

$G_I$—interlayer tension, mode II—$G_{II}$—sliding shear, mode III—$G_{III}$—scissoring shear [32]. The rate of energy release is determined by the equation (Figure 1):

$$W_{\text{closing}} = \frac{1}{2}F_{2,5}v_{2,5} = \frac{1}{2}F_{2,5}v_{1,6}$$

$$G = \frac{W_{\text{closing}}}{\Delta A}$$

where $\Delta A = \Delta db$—crack area increment, $\Delta d$—length of elements at the crack tip, $b$—element width, $F_{i,j}$—forces at the crack tip, $v_{i,j}$—displacement of crack tip nodes.

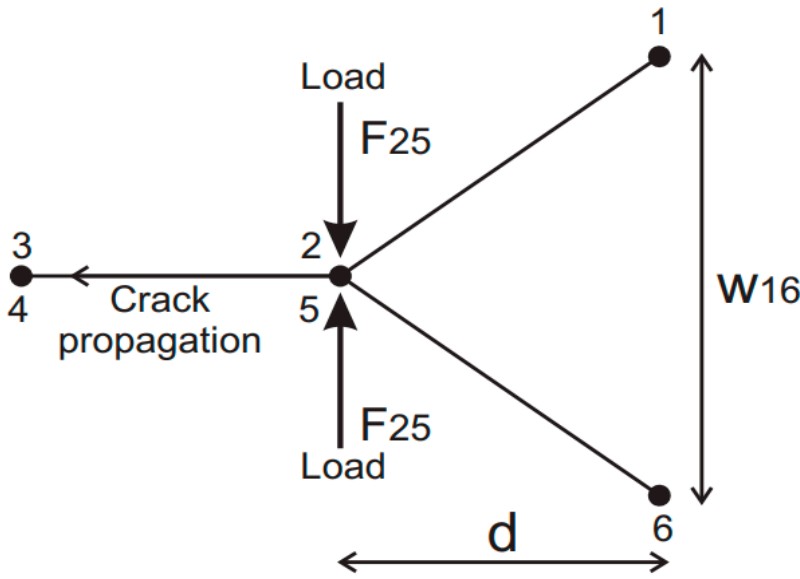

**Figure 1.** To VCCT explanation [30].

*3.2. Model Description*

To obtain $G(a)$ dependency, FE models of a composite panel with damage zones of different sizes were constructed. The choice of the panel as an object of study is associated with the prevalence of such composite structures in modern aircrafts.

The geometric parameters of the panel are presented in Table 1 and Figure 2. The selected geometric parameters correspond to the dimensions of the samples used to study the residual compressive strength after impact D7137 [34]. The layers of the panel laminate were given the properties of a typical aircraft unidirectional CFRP (Table 2). To model CFRP, linear-elastic orthotropic model material was used. CFRP properties were defined as Engineering Constants.

The FE model of the panel is shown in Figure 3. Panels were made with a ply-by-ply modeling approach to obtain the failure modes encountered during the testing of composite panels, delamination especially, so the layers of the panel were modeled with three-dimensional plates built from 60,000 solid C3D8 elements, and the number of nodes in the model was 120,000. In the regular zone, the element size was 2 × 2 mm, while in the damage zone, the mesh was refined with bias to the damage zone boundary with a size of 0.2 × 0.5 mm to obtain correct values of the energy release rate.

**Table 1.** Panel geometric parameters.

| Parameter | Value |
|---|---|
| Width, W, mm | 100 |
| Length, H, mm | 150 |
| Damage zone size, a, mm | 6, 7.4375, 8.875, 10.3125, 11.75 |

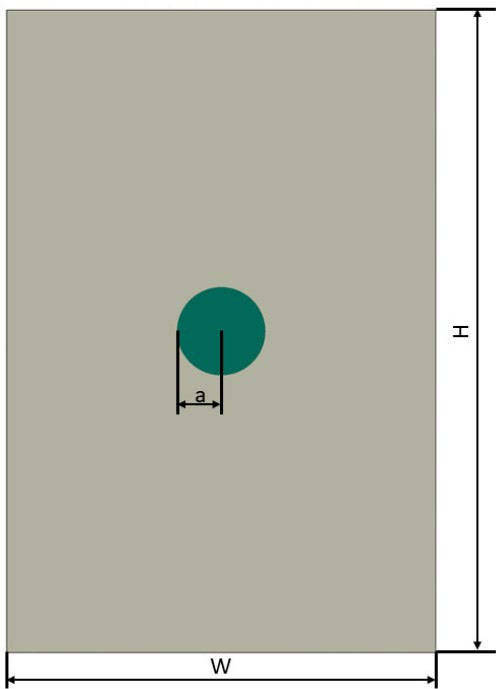

**Figure 2.** Model general view.

**Table 2.** T300/5208 properties [35].

| Parameter | Value |
|---|---|
| Elastic modulus under longitudinal tension, $E_1$, GPa | 138 |
| Elastic modulus under transverse tension, $E_2 = E_3$, GPa | 15 |
| Poisson's ratio, $v_{12} = v_{13} = v_{23}$ | 0.21 |
| Shear modulus, $G_{12}$, GPa | 5.9 |
| Shear modulus, $G_{13} = G_{23}$, GPa | 5.9 |
| Ultimate longitudinal tension strength, $F_{1t}$, MPa | 1420 |
| Ultimate longitudinal compression strength, $F_{1c}$, MPa | 1320 |
| Ultimate transverse tensile strength, $F_{2t}$, MPa | 43.1 |
| Ultimate transverse compression strength, $F_{2c}$, MPa | 160 |
| Shear strength, $F_{12}$, MPa | 112 |
| Monolayer thickness, $t$, mm | 0.127 |

In each model, the layers were assembled into a lay-up $[45, -45, 0_3, 90]_s$, the total panel thickness $t = 1.397$ mm. This lay-up was chosen for the study since it is a typical lay-up that is used in industry (for example, aircraft wing panels) and contains all possible interlayer interfaces (except [45, 90]).

To implement an interface failure between layers, the surface-based cohesive zone method (CZM) [32] was used, which was set between all interfaces, except the interface with the damage zone. The possibility of cohesive zone failure, which led to delamination between the layers, was realized in accordance with the theory of fracture mechanics with the use of the maximum stress criterion. While damage initiation criteria were not satisfied, plies were bonded together. The strength properties of the cohesive zone correspond to the strength of the matrix and are presented in Table 3. The stiffness of CZ was adopted by default. To define CZ, corresponding surfaces of plies were selected.

The damage zone was placed between layers 2 and 3 (layers with orientation $-45$ and 0). In this interface, instead of a cohesive zone, surface-to-surface contact with VCCT was used. In the damage zone, there was no connection between the nodes of the layers, while the remaining nodes of the interface were connected to each other. VCCT parameters are presented in Table 4. It was decided that the values of ERR taken for further analysis

of crack growth correspond to mean values of the densest zone with maximum ERR values. From further investigation, it appeared that these zones situate on the sides of the delamination and are normal to the applied load direction. The local peaks of ERR, which might be attributed to mesh and contact algorithm effects, were not taken into account in the analysis.

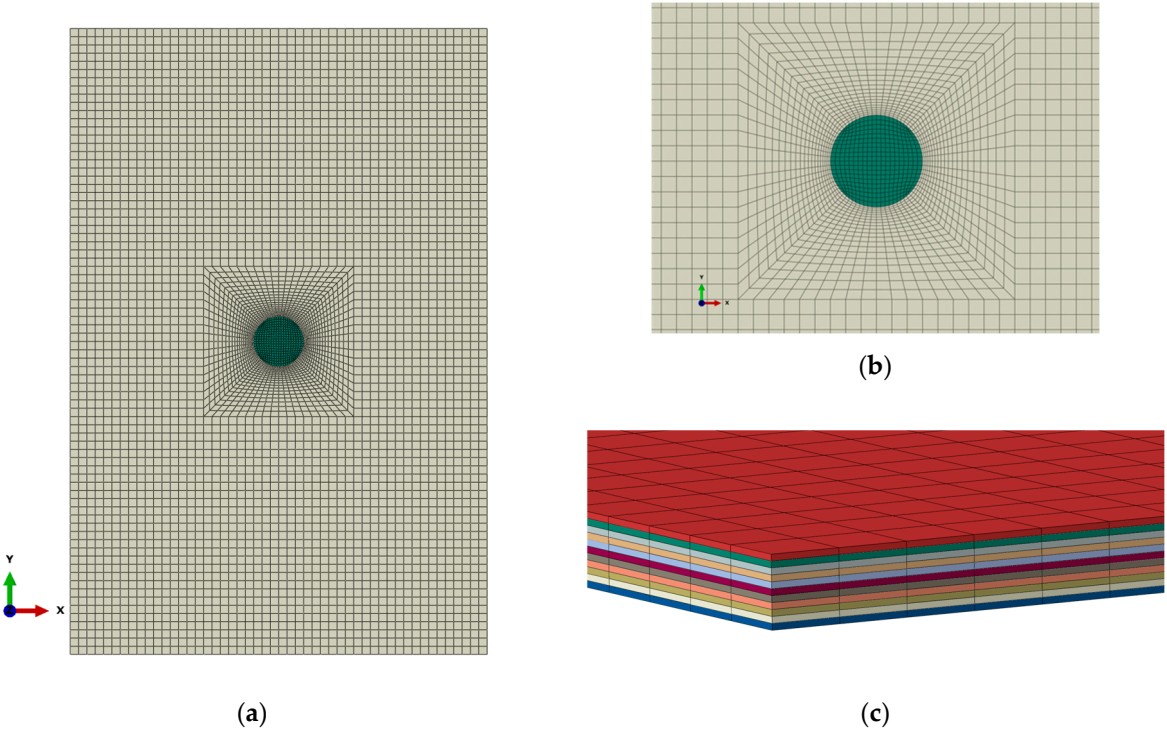

**Figure 3.** (**a**) FEM general view; (**b**) damage zone FE mesh; (**c**) ply-by-ply modeling.

**Table 3.** Maximum stress criteria for CZM.

| Parameter | Value |
|---|---|
| Normal stress, $\sigma_n$, MPa | 40 |
| Shear-1 stress, $\tau_1$, MPa | 90 |
| Shear-2 stress, $\tau_2$, MPa | 90 |

**Table 4.** VCCT parameters.

| Parameter | | Value |
|---|---|---|
| ERR | mode I, $G_I$, $\frac{MJ}{mm}$ | 0.28 |
| | mode II, $G_{II}$, $\frac{MJ}{mm}$ | 0.28 |
| | mode III, $G_{III}$, $\frac{MJ}{mm}$ | 0.28 |
| Exponent, n | | 2.284 |

The following boundary conditions were applied to the panel: clamping alongside 1 and clamping along sides 2 and 3 with the possibility of moving along the load axis (Figure 4). A displacement was applied to side 4. These boundary conditions correspond to those implemented in compression after impact tests [34]. Load introduction simulates one-way load excursion in compression.

The Abaqus/Explicit finite element solver was used, which uses an explicit scheme for solving the dynamic problem. In order to implement the quasi-static solution, the displacement to side 4 was applied at such a speed that the kinetic energy of the system was <5% of the strain energy.

The parameters of the damage growth rate equation are taken from [28], $c = 0.0275$, $\beta = 3.1$.

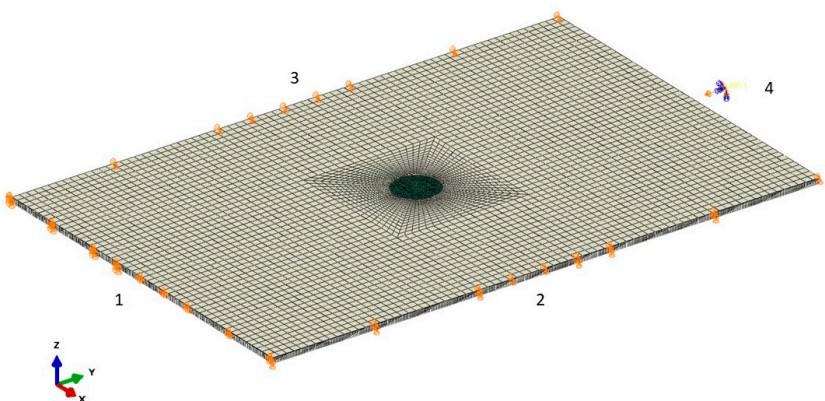

**Figure 4.** Boundary conditions.

### 3.3. ERR Calculation

With the help of the finite element models shown above, the analysis of models with different sizes of damage zones was carried out according to the design cases presented in Table 5.

Based on the results obtained for each design case, the energy release rate was determined at a force equal to 60% of the failure load for a model with a minimum damage zone size. The value of the cyclic load was taken on the basis of the data in [36] and represents the value of the level of operational load of a typical flight [37].

**Table 5.** The design cases for CZM.

| Design Case | Damage Zone Size |
|:---:|:---:|
| 1 | 6 |
| 2 | 7.4375 |
| 3 | 8.875 |
| 4 | 10.3125 |
| 5 | 11.75 |
| 6 | 15 |

## 4. Results and Discussion

The panels were compressed up to failure, due to which failure loads were obtained depending on the damage size. In all design cases, the following failure sequence was implemented. At first, the panel absorbs the compression load as an intact panel, but when a certain level of loads was reached, local buckling began in the damage zone: (Figure 5). The complete loss of the panel bearing capacity is associated with the buckling of the entire panel (Figure 6). In all calculations, stresses in the stress concentration zone near the damage boundary did not reach the layer strength (Table 2).

In the results of the numerical simulations of laminated plates with different lamination sizes, the $G(a)$ dependency was developed (Figure 7), which is necessary for damage growth period evaluation, along with a dependency of failure load versus damage size (Figure 8), with the help of which the critical damage size choice is motivated and used as a final damage size in the expression for the damage growth period.

The obtained relation $G(a)$ for the level of load, equivalent to the typical flight spectrum [36,37], at first, helps to recognize a threshold value of defect size $a_{th}$, at which the rate of $G(a)$ starts to increase. As it is seen from the figure, $a_{th}$ literally coincides with the chosen starting defect size (BVID) and equals 5.8 mm. This value might be chosen as the threshold of detectability value, so that it can be recognized using standard methods of NDE in service, such as ultrasound inspection and shearography. Then, on plot $G(a)$, three

regions of damage growth are clearly observed—initially, up to ≈ 9 mm with the average rate of $G$ growth, it is replaced by the period of nearly constant $G$ in the range of the defect sizes from 9 to 12 mm, and the final period of $G$ growth intensification with respect to the first two regions. It is worth noting that at no region does the ERR $G$ reach the levels compared with static fracture toughness, and up to the final region on the curves, the levels of ERR are at least one order of magnitude lower than static fracture toughness.

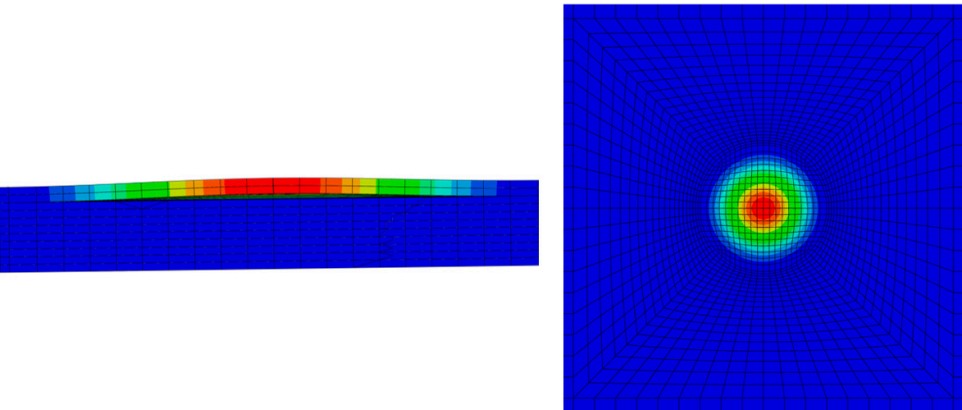

**Figure 5.** Delaminated layers local buckling.

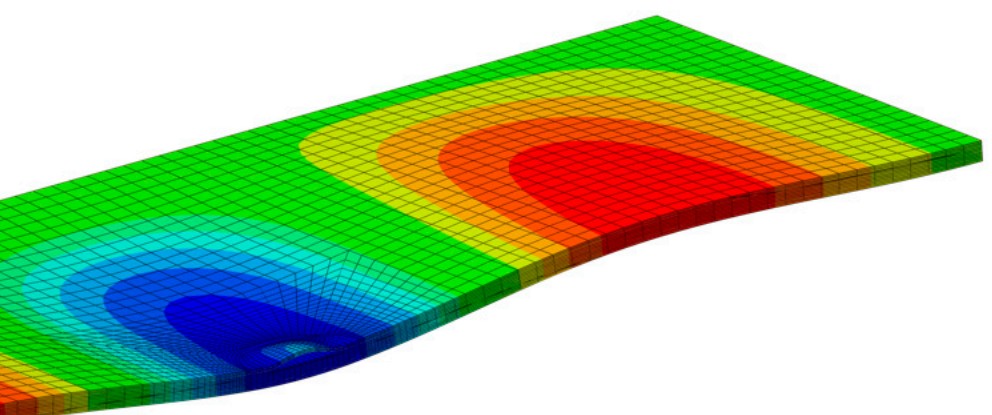

**Figure 6.** Panel buckling.

It was mentioned in the methodology description that for the damage growth period derivation, the total ERR $G_{tot}$ is used, but for the sake of understanding the mechanical nature of impact damage growth, it is essential to investigate the impact of each delamination mode in total ERR. From Figure 7, it is seen that for all regions of damage sizes, the $G_I$ and $G_{II}$ have the same magnitude and equally contribute to $G_{tot}$, which coincides with usual assumptions in different research [25,26]. Along with this, the trends of $G_I$ and $G_{II}$ are different: $G_{II}$ monotonically increases during all the stages identified; meanwhile, $G_I$, and hence $G_{tot}$, has a different trend depending on the size of the damage. On the third step of damage increase, separating mode $G_I$ becomes dominant. The tearing mode $G_{III}$, which is always neglected [25,26], is nearly two times smaller than $G_{II}$, and similarly has a monotonical increase with a slope smaller than sliding mode $G_{II}$. In the present study, total ERR $G_{tot}$ is obtained using $G_{III}$ because it shows an obvious influence and due to the fact that it used the form and constants of damage rate equations and is likely to utilize all modes for $G_{tot}$ (1) [28]. In this way, the mechanism of damage propagation under the buckled substrate includes the action of both separating direct stresses due to loss of stability and transverse shear stresses on the boundary of the damage zone.

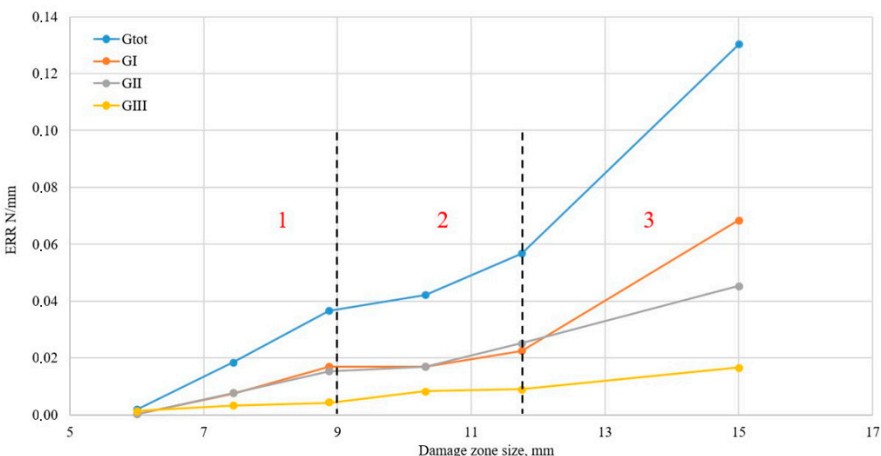

**Figure 7.** Energy release rate versus damage size dependency.

The analysis of gained relation in Figure 8 suggests the choice of critical damage size $a_f$, being such a size, at which the failure load of the panel $P_f$ becomes lower than the limit load level. For load-bearing metallic aircraft structures where the damage growth is anticipated reliably [38], and for composite structures with visible damage [2], the limit load level equals 67% of the ultimate load. From Figure 8, it is evident that such a level corresponds to the size $a_f = 11.75$ mm.

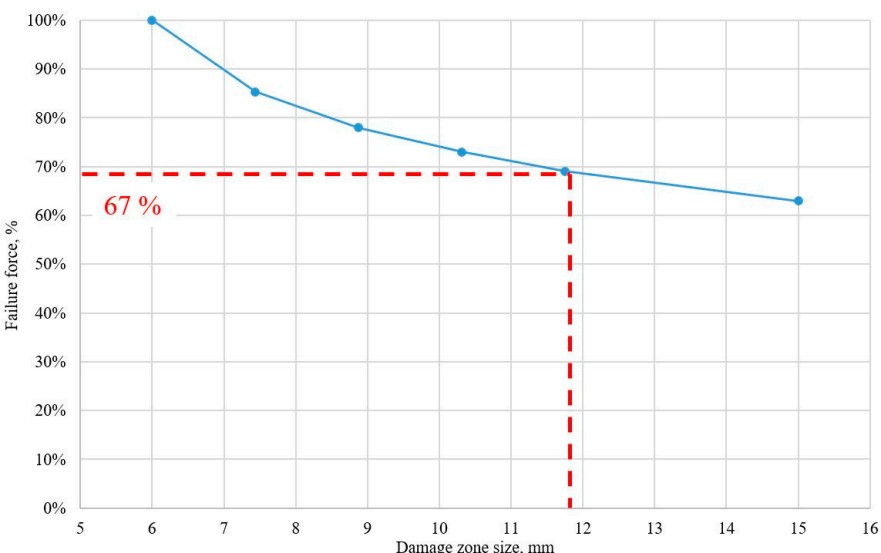

**Figure 8.** Failure load against damage size.

After the derivation of critical damage size and using trends of $G(a)$, it becomes possible to obtain the damage growth period $\Delta N$ using Equation (1). With the designations explained in Figure 8, the procedure for $\Delta N$ derivation is conducted as follows:

$$\Delta N = \Delta N^1 + \Delta N^2 + \Delta N^3$$

$$\Delta N^i = \int_{a_{0,i,i}}^{a_{1,i}} \frac{da}{c(k_i(a - a_{th,i}))^\beta} = \frac{1}{k_i c} \frac{1}{(1 - \beta)} \left( G_{tot1,i}^{(1-\beta)} - G_{tot0,i}^{(1-\beta)} \right) \qquad (2)$$

$$i = 1, 2$$

where $i$—is a number of regions in $G(a)$ curve, $k_i$—$G(a)$ curve slope on region $i$. The rest of the designations are illustrated in Figure 9.

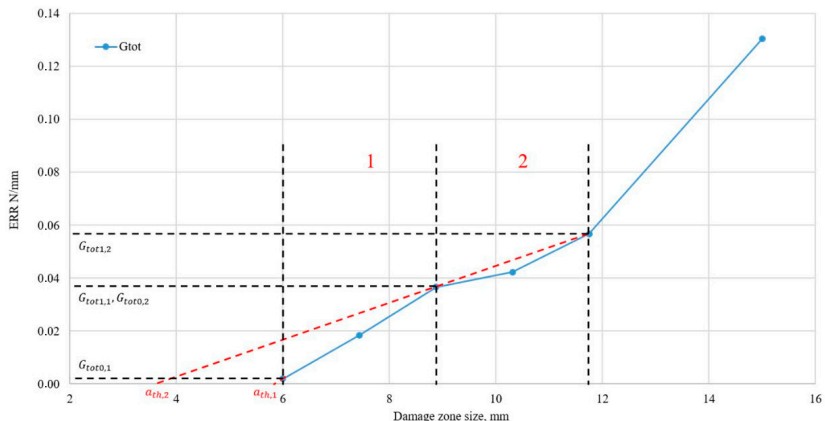

**Figure 9.** Designations used in final form of damage growth life Equation (2).

The results of the calculations are collected in Table 6.

**Table 6.** Damage growth life results.

| Region | Initial Damage Size, $a_0$, mm | Final Damage Size, $a_1$, mm | Damage Growth Period, $\Delta N$, Cycles |
|---|---|---|---|
| 1 | 6 | 8.875 | $7.262 \times 10^8$ |
| 2 | 8.875 | 11.75 | $1.548 \times 10^6$ |
| 1 + 2 | 6 | 11.75 | $7.278 \times 10^8$ |

From the Table 6, it is obvious that with literally the same increments of damage size in the 1st and 2nd regions (Figure 7), the main part of the resulting damage growth life is contributed by the first period, which is a direct consequence of the fact that the total ERR $G_{tot}$ is significantly larger all along the region, even if it does not have such a gradual increase with damage size growth, as in the first region. In this way, it might be concluded that the magnitude of crack driving force (in this case—$G_{tot}$) is a parameter with a more pronounced influence on the damage tolerance of the structural element than the rate at which it grows with respect to the size of the damage.

## 5. Conclusion and Future Works

The proposed semi-analytical method of analysis of BVID growth under cyclic compression in flat composite panels allows for an effective estimate of the damage growth period, which is required to ensure the damage tolerance of a typical composite element of the aircraft structure. Analytical estimates were conducted in light of the introduced concept of reference damage mode. This hypothesis must be proved by means of direct experimental study. The obtained dependency of damage driving force against the size of the damage does not naturally suit the application of the slow-growth design principle in the typical composite element of aircraft structure because the damage growth rate does not slow down along with damage size increase [24], but still suits for the purpose of critical damage size choice. In this way, the critical damage size matches the one for which the failure load of the structure reaches the limit load to be under the existing regulations [2,39]. The case study conducted advocates in an illustrative manner the applicability of slow-growth ideology for composite structures subjected to cyclic compression service. However, the derived damage growth periods must be refined in the future by choosing the appropriate physical measure of damage driving force for the case of cyclic compression and by getting the parameters of the damage growth rate equation from the special experiment.

**Author Contributions:** Conceptualization, N.T. and K.S.; methodology, N.T.; software, N.T. and K.S.; validation, N.T. and K.S.; formal analysis, N.T. and K.S.; investigation, N.T. and K.S.; resources, N.T. and K.S.; data curation, N.T.; writing—original draft preparation, N.T. and K.S.; writing—review and editing, N.T. and K.S.; visualization, N.T. and K.S.; supervision, Turbin, N; project administration, Turbin, N; funding acquisition, N.T. All authors have read and agreed to the published version of the manuscript.

**Funding:** The article is prepared in the implementation of the program for the creation and development of the World-Class Research Center "Supersonic" for 2020-2025 funded by the Ministry of Science and Higher Education of the Russian Federation (Grant agreement of 20 April 2022 No 075-15-2022-309).

**Institutional Review Board Statement:** Not applicable.

**Informed Consent Statement:** Not applicable.

**Data Availability Statement:** No new data were created.

**Conflicts of Interest:** The authors declare no conflict of interest.

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
