# Peer review of "Analysis Method for Post-Impact Damage Development in Carbon Fiber Reinforced Laminate under Repeated Loading"

_jcs, doi:10.3390/jcs7050201_

Round 1
Reviewer 2 Report
- In line 28, a parenthesis is missing;
- Throughout the text there are several missing references.
- In line 166, the authors stated that the Load Ratio for this particular problem is infinite. However, as demonstrated in “A. Raimondo, C. Bisagni. Analysis of Local Stress Ratio for Delamination in Composites Under Fatigue Loads, AIAA journal, Vol. 58 (1), 2020”, a distinction must be made between the applied load ratio and the local stress ratio, the ratio between the minimum and maximum stress at the crack tip, which actually contribute to the opening of the delamination. These two values may be different. The authors should comment this point in the text.
- Regarding the boundary conditions in Figure 4, it is not clear how the load is applied, is only the maximum load considered or is the entire fatigue cycle simulated?
- Figure 7 reports the Energy Release Rate, however it is not clear in which points along the circular delamination front it is evaluated. Is it the maximum value?
- Figure 7 is not cited in the text.
- Regarding the cohesive zone method, what kind of approach has been adopted: Cohesive elements or cohesive surfaces?
Round 2
Reviewer 1 Report
Accept in present form
Reviewer 2 Report
The authors have revised the manuscript according to the previous reviewers' comments, hence the paper is recommended for publication.